# Modelling Spatiotemporal Patterns of Lyme Disease Emergence in Québec

**DOI:** 10.3390/ijerph18189669

**Published:** 2021-09-14

**Authors:** Marc-Antoine Tutt-Guérette, Mengru Yuan, Daniel Szaroz, Britt McKinnon, Yan Kestens, Camille Guillot, Patrick Leighton, Kate Zinszer

**Affiliations:** 1Lady Davis Institute for Medical Research, 3755 Chemin de la Côte-Sainte-Catherine, Montréal, QC H3T 1E2, Canada; ma.guerette@ladydavis.ca; 2Centre de Recherche en Santé Publique, 7101 Av du Parc, Montréal, QC H3N 1X9, Canada; mengru.yuan@mail.mcgill.ca (M.Y.); daniel.szaroz@umontreal.ca (D.S.); britt.mckinnon@umontreal.ca (B.M.); yan.kestens@umontreal.ca (Y.K.); camille.guillot@umontreal.ca (C.G.); patrick.a.leighton@umontreal.ca (P.L.); 3Département de Médecine Sociale et Préventive, École de Santé Publique, Université de Montréal, 7101 Av du Parc, Montréal, QC H3N 1X9, Canada; 4Research Group on Epidemiology of Zoonoses and Public Health (GREZOSP), Faculté de Médecine Vétérinaire, Université de Montréal, 3200 Rue Sicotte, Saint-Hyacinthe, QC J2S 2M2, Canada; 5Department of Epidemiology, Biostatistics and Occupational Health, McGill University, 1020 Pine Ave W, Montréal, QC H3A 1A2, Canada

**Keywords:** Lyme disease, Québec, spatiotemporal patterns, front wave velocity, clusters, emergence

## Abstract

Lyme disease is a growing public health problem in Québec. Its emergence over the last decade is caused by environmental and anthropological factors that favour the survival of *Ixodes scapularis*, the vector of Lyme disease transmission. The objective of this study was to estimate the speed and direction of human Lyme disease emergence in Québec and to identify spatiotemporal risk patterns. A surface trend analysis was conducted to estimate the speed and direction of its emergence based upon the first detected case of Lyme disease in each municipality in Québec since 2004. A cluster analysis was also conducted to identify at-risk regions across space and time. These analyses were reproduced for the date of disease onset and date of notification for each case of Lyme disease. It was estimated that Lyme disease is spreading northward in Québec at a speed varying between 18 and 32 km/year according to the date of notification and the date of disease onset, respectively. A significantly high risk of disease was found in seven clusters identified in the south-west of Québec in the sociosanitary regions of Montérégie and Estrie. The results obtained in this study improve our understanding of the spatiotemporal patterns of Lyme disease in Québec, which can be used for proactive, targeted interventions by public and clinical health authorities.

## 1. Introduction

Lyme disease (LD) was first identified in Lyme, Connecticut in 1982 in response to several years of unusually high rates of arthritis in both children and adults [1]. Since this time, Lyme disease has been detected in several countries in the Americas, Europe, and Asia [2]. The global incidence is difficult to determine because of varying levels of awareness, surveillance, and reporting throughout the world; however, there are up to 567,000 estimated new cases documented each year, and the US reported approximately 476,000 new diagnoses in 2018, making LD a significant global health challenge, particularly in North America [3,4,5]. Mainly found in the United States, the majority of cases are concentrated in the northeastern states [6]. In Canada, the disease was first detected in the 1980s in southern Ontario with most current cases occurring in Ontario, Québec, and Nova Scotia [7]. Lyme disease infection can result in significant illness including musculoskeletal, cardiac, and neurological disorders, with the most common sequelae being arthritis [8,9,10]. Chronic musculoskeletal and neurological disorders of Lyme infection have been reported to persist in approximately 10–13% of patients diagnosed with the disease, and in the United States up to 32% of patients develop arthritis [11,12,13].

The Lyme disease-causing bacterium in Canada and in the United States is *Borrelia burgdorferi*, which is transmitted to animals and humans by the black-legged tick (*Ixodes scapularis*) in eastern North America, and the western black-legged tick (*I. pacificus*) on the west coast of North America [14]. The geographical range expansion of *I. scapularis* in eastern North America is largely driven by long range movement of tick reservoirs such as migratory songbirds, as well as short range spread by white-tailed deer, *Odocoileus virginianus*, and the white-footed mouse, *Peromyscus leucopus*, as well as other mammals [15,16]. In Québec, the white-footed mouse is more frequently found in the southwestern part of the province [17], whereas white-tailed deer are abundant throughout the deciduous forests of southern Québec and less so in the boreal forest (Figure 1) [18]. Blacklegged tick mortality is largely influenced by temperature and as temperatures drop, mortality increases [19]. Due to warming trends in Québec, the survival rate of ticks has been increasing as weather extremes become more common and yearly heat records are documented in the southern part of the province [20,21,22].

Environmental fragmentation influences the ecological characteristics of the white-footed mouse and the white-tailed deer. In southern Québec, considerable development in the agricultural sector over the last decades has led to increased logging practices [23,24,25]. This fragmentation decreases mammalian and plant biodiversity, while increasing the density of tick reservoirs [26,27,28,29]. The white-footed mouse is a species highly influenced by these changes and tends to cluster in transitional zones, spreading rapidly in southern Québec where high-density populations are found in these fragmented regions [26,28,30,31]. The risk of human exposure to ticks in these types of areas is greater, both in terms of the dispersion of its reservoirs but by also through more frequent contact related to recreational or professional human activities [32,33].

Several spatiotemporal studies on Lyme disease have been conducted in North America, mainly exploring the distribution of the tick and its reservoirs, but studies on the distribution of human cases in Canada remain limited [34]. The distribution of *I. scapularis* in Canada is restricted mainly to southwestern Québec, southeastern Manitoba and Ontario, and the Maritime provinces, particularly in Nova Scotia and some in New Brunswick, which coincide with observed clusters of human Lyme disease cases in those regions [9,29,34,35,36,37,38]. The spatiotemporal emergence of *B. burgdorferi* and its vector and reservoirs have been examined in Québec, but the spatiotemporal patterns and the velocity of human cases of Lyme disease emergence have not been studied. The objectives of this study were to estimate the direction and speed of human Lyme disease emergence in Québec and to characterize the spatiotemporal patterns of the emergence.

## 2. Materials and Methods

### 2.1. Data

Data on cases of Lyme disease reported in Québec between 1 January 2004 and 31 December 2018 were obtained from the Québec National Institute of Public Health (INSPQ) with regional public authorisation. Cases reported between 2004 and 2005 were either acquired outside the province or had unknown locations of exposure and were excluded. Lyme disease case data in Québec are collected by the reportable diseases registry and Lyme disease has been a notifiable disease since 2003. After receiving confirmation of a new case of Lyme disease, public health authorities undertake an epidemiological investigation to identify the likely sites of exposure [39]. The variables required for the analyses were individual-level, such as the date of notification, date of symptom onset, municipality of acquisition, as well as age and sex. Population data by municipalities between 2001 and 2018 were downloaded from the Québec Institute of Statistics [40]. Québec municipality shapefiles were downloaded from Statistics Canada [41]. The centroids of each municipality, i.e., the coordinates that correspond to geographic centres, were plotted using the open-source geographic information system QGIS 3.4.2. Data management was done with R 3.5.2.

### 2.2. Surface Trend Analysis

A surface trend analysis was modelled to estimate the front wave velocity of Lyme disease emergence in Québec, using the R package “outbreakvelocity” [42]. The first notified case of Lyme disease for each municipality was identified. This was then converted into time in weeks between the first ever reported case of Lyme disease in Québec and each first case per municipality. The centroid coordinates of each municipality, in metres, were also included.

The velocity of emergence was calculated by estimating polynomial regression models between time and the *x* and *y* centroid coordinates [43,44]. A continuous surface of time to emergence was first estimated by least squares in a linear two-dimensional model. As a simple 2-dimensional plane through the points was insufficient to model the data, high-order polynomials were explored to capture local scale trends. We explored 6 models by incrementally adding polynomials up until the order 6 and the best fit model was selected using the AIC and BIC values. The model with polynomial terms of order 3 provided the best fit:$$f\left(t|x,y\right)=\beta_{0}+\beta_{1}X+\beta_{2}Y+\beta_{3}X^{2}+\beta_{4}Y^{2}+\beta_{5}X^{3}+\beta_{6}Y^{3}+\beta_{7}\mathrm{XY}+\varepsilon$$
where E(ε)=0.

The vectors were converted to express the magnitude and direction of rate of change (in kilometers per year) by finding the inner product of the vector, where magnitude ||xy|| = √(x^2^ + y^2^) and the direction θ = tan − 1 (y/x).

The analysis was conducted separately using the date of symptom onset and the date of notification, given the difficulties in establishing a reliable symptom onset date for Lyme disease. We estimated the average speed of Lyme disease emergence in Québec as well as the velocity relative to northern spread. A sensitivity analysis was then conducted to assess velocity changes according to the INSPQ definition of Lyme disease risk regions (municipalities with two or more reported cases in the last five years) [45] and for municipalities with two or more reported cases throughout the study period. This was done using the first declared case for each municipality having declared at least two cases (Appendix A). Overall, six surface trend analyses were conducted, as described in Table 1.

Average velocity of human case emergence was calculated by taking the mean speeds and directions of each vector of spread, as represented by *x* and *y* coordinates, across all municipalities where Lyme disease was documented. The average northern spread of Lyme disease was then estimated by *y* coordinates alone to represent 0° to the North.

### 2.3. Cluster Analysis

The SaTScan software was used to detect statistically significant clusters of Lyme disease in Québec from 2004 to 2018 for all municipalities having reported at least one Lyme disease case during the study period. SaTScan uses a circular window over the study region with a user-defined radius and across time and/or space, to calculate the difference between observed and expected cases [46]. We used a discrete Poisson model to identify spatiotemporal and spatial clusters that could overlap. The analysis was conducted using both the date of notification and the date of symptom onset.

For this analysis, the total number of cases per municipality per year were included along with the average population per municipality over the study period and the centroids of each municipality. The temporal unit was specified in years as Lyme disease incidence is generally low in Québec. We specified the cluster size to be no more than 15% of the Québec population at risk, within circles with a maximum radius of 25 km. This was chosen as we did not want to identify overly large clusters that may overlap non-endemic areas [47]. A relative risk (RR) was estimated for each significant cluster, which provided the risk of LD within the cluster relative to the risk outside the cluster. The RR is calculated by dividing the observed cases with the expected cases within the cluster, the result of which is then divided with the observed cases divided by the expected cases outside the cluster, as represented by the following formula:$$\mathrm{RR}=\frac{c/E\left[c\right]}{\left(C-c\right)/\left(C-E\left[c\right]\right)}$$
where *c* = observed cases within the cluster, *E*[*c*] = expected number of cases, and *C* = total number of cases in the dataset. The clusters were then mapped using QGIS 3.4.2.


## 3. Results

From 2006 to 2018, there were 727 confirmed or probable cases of Lyme disease from 138 of 1476 municipalities in Québec. Cases reported between 2004 and 2005 were either acquired outside the province or had unknown locations of exposure. The total number of annual cases reported between 2006 and 2012 was low (between 1 and 13 cases reported each year), with the first locally acquired cases appearing in the municipality of Coaticook in Estrie. Being geographically near the border, it is probable that these cases were imported from the US by migrating tick reservoirs. There was a significant increase in the number of cases in 2013, 2015, and 2017 relative to their previous years. The highest observed number of cases was in 2017 with 212 reports, more than twice the number in 2016 (Figure 2).

Nearly 71% of cases were acquired in the regions of Montérégie and Estrie. The municipality with the highest number of cases was Bromont in Estrie (*n* = 138), while most of the affected municipalities (71%) had fewer than 10 cases (Figure 3). There is a bimodal distribution of cases amongst those aged 1–19 and 50–69, while the latter represented the highest reporting frequency (Table 2). Men also accounted for 57% of all cases and were statistically more at risk than women of being exposed to Lyme disease (RR = 1.34 [CI_95_: 1.16–1.55]). Cases were frequently reported several weeks, and in some cases years, after the onset of symptoms. The average difference between the appearance of symptoms and medical consultation was two months, with a standard deviation of 148 days. The date of symptom onset was missing for 34 cases. Cumulative number of Lyme disease cases by date of symptom onset are presented in the Appendix A.

### 3.1. Surface Trend Analysis

For all six surface trend analyses, the best fit model was a third order polynomial linear model. Results of the analyses for all LD cases reported over the entire study period based on symptom onset date and notification date are described in the main text. Detailed information on the remaining four analyses (A2, A3, B2, B3) are included in Appendix A, and a summary of all six models is presented in Table 3.

The estimated velocity of human Lyme disease spread was northward for both model A1 (Figure 4) and B1 (Figure 5). The speed of northern emergence was more rapid in the model using the date of symptom onset, estimated at 32 km/year, and 18 km/year in the model using the date of notification. For the majority of municipalities, the speed of Lyme disease introduction was less than 50 km/year. In general, the speed of emergence increased the further north a municipality was located, as seen in many neighboring the St-Lawrence River, with speeds varying between 50–100 km/year. Emergence was particularly accelerated in municipalities located within the administrative regions of Outaouais, Lanaudière, and the Laurentians, varying between 100–600 km/year. There was an outlying municipality with a speed of emergence of 2950 km/year in Beauceville, in the Chaudière-Appalaches administrative region as estimated by model B1.

### 3.2. Cluster Analysis

Spatiotemporal and purely spatial clusters were identified for each case definition, with the purely spatial clusters presented in the Appendix A. A spatiotemporal analysis was conducted separately based on the date of symptom onset and for date of notification for all municipalities having reported at least one case over the entire study period. Seven significant clusters of Lyme disease were detected for both dates. As the clusters for both dates were identical in time and space, with little variance between the two, only spatiotemporal clusters based on the date of notification are presented (Figure 6 and Table 4), with spatiotemporal clusters based on date of symptom onset detailed in the Appendix A. The clusters were located in southwestern Québec, in the Montérégie, Estrie, and Centre-du-Québec regions. The relative risk between the different identified clusters ranged from 97.8 to 10.5 and differed according to timeframe (2016–2018 vs. 2013–2018). The three clusters bordering the USA represent the earliest emergence of Lyme disease cases in Québec with clusters emerging further north with time.

## 4. Discussion

Human cases of Lyme disease are emerging in Québec and its spread has been largely northward over the past decade. The speed of spread varied between 18 and 32 km/year, depending on the use of notification versus symptom onset date. The disease is primarily found in southwestern Québec with significant clusters in Montérégie, Estrie, and Centre-du-Québec, with the city of Bromont and surrounding area being most at risk for LD in the province. The direction of emergence indicates that several municipalities further north in the administrative regions of Outaouais, Laurentians, and Mauricie are at risk of LD becoming endemic.

The risk of acquiring Lyme disease was significantly elevated in seven clusters that include 64 municipalities in southwestern Québec, where outdoor leisure and tourism is popular. Three of these clusters have been at-risk since 2013, which coincides with the significant increase in provincial LD notifications. These three clusters also border the United States, where Lyme disease has been a major public health issue in the northeastern states since the 1980s [1,6]. The appearance of cases in Québec is largely due to the migration of ticks in relation to bird migration, particularly in more remote communities in the province, but also connected to climatic changes and forest fragmentation that drive the ecological patterns of tick reservoirs such as the white-footed mouse and increase tick survival rates [26,35,48,49]. The four more recent clusters are found further north and became at-risk in 2015 and 2016. The northward trend of LD is consistent across Canada, including in Manitoba, Ontario, New Brunswick, and parts of Nova Scotia, but most noticeably in eastern Ontario, where rapidly invading populations of infected *I. scapularis* have been followed by a continuously increasing incidence of human Lyme disease [50,51,52]. Clusters of infected *I. scapularis* are found in southwestern Québec in high densities, with their dispersion overlapping with the clusters identified in our study [38]. The rate of *B. burgdorferi* infection was minimal for ticks collected between 2004 and 2010, which may explain the small number of cases reported during this period [16,38,53].

The spread of the *B. burgdorferi* pathogen by its vector *I. scapularis* is primarily driven by migratory birds in Québec, especially when considering Lyme disease cases have been declared north of the St-Lawrence River, which would act as an important dispersal barrier to land-dwelling hosts such as the white-footed mouse [54,55]. However, these rodents are sensitive to environmental changes such as rising temperatures and forest fragmentation which increase favourable habitats and winter survival rates while spreading the pathogen at a more local scale [26,49]. While these environmental determinants are not necessarily mutually exclusive from one another, these factors both influence dispersal independently. Albeit framed in a short timescale relative to climate change, temperature averages have been warming for the past 60 years in Québec [20], and multiple studies have shown that climate change is an important driver of pathogen spread in the northeastern regions of the United States, where Lyme disease has been endemic since the 1980s [56,57,58]. It has also been estimated that the speed of emergence of *B. burgdorferi* in Québec is between 3.5 and 11 km/year [26] while the rate of introduction of the white-footed mouse is 10 km/year in the province [49]. The speed of *I. scapularis* invasion has not been estimated specifically for the province of Québec, but a nation-wide study has modelled an average expansion of 46 km/year [16]. Our study estimates a faster rate of spread of human Lyme disease compared to its ecological reservoirs in Québec, which is plausible considering the role of long-range spread by migratory birds. Human cases may thus appear in more northern municipalities before tick-carrying mice and deer appear in these regions. Long-range spread by birds could also explain the high estimated speeds of human LD emergence in regions north of the St-Lawrence River, as reported cases were declared in isolated regions far from one another in a short temporal scale. This in turn may skew our estimates towards faster spread relative to that of tick reservoirs. Transmission of *B. burgdorferi* to humans may also occur at different rates, based on the probability of contact between susceptible individuals and infected ticks [59].

We observed a bimodal pattern of cases for 1–19 and 50–69-year-old age groups, the latter representing the age group most at risk. Men were also statistically more at risk than women. A similar demographic trend has been documented in the United States, where cases of Lyme disease are more prevalent among men aged 5–15 years and 45–55 years [60]. In Europe, women are at higher risk with a bimodal distribution of cases for children and older adults [60,61]. In Québec, the agriculture and forestry sector employed 2.6 times more men than women in 2017, which may explain the higher risks associated with men [62,63]. Furthermore, a 2017 study on the participation rate of outdoor activities in Québec shows that 41% of men versus 35% of women regularly participate in outdoor activities [64]. However, household yards are an important exposure source of Lyme disease, and factors such as pet ownership, bird feeders, and outdoor dining areas significantly increase the risk of private household owners of finding ticks on themselves [65,66,67]. It has also been shown that children are at high risk of Lyme disease, because they are at increased risk for bites from nymphal ticks that are harder to find on the body, and also less likely to have ticks removed in time to prevent transmission [68,69,70]. More severe symptoms can occur up to several months after the initial exposure to an infected tick, resulting in a significant lag between when an individual was infected to when they seek medical care. On average, there was an 8-week delay between onset date and registration date.

In Canada and Québec, Lyme disease is a current public health problem and, despite a better level of awareness in recent years, it remains a relatively unappreciated issue [71]. It is very likely that the actual number of cases of Lyme disease is underrepresented in our study, attributable to a lack of general knowledge amongst health professionals and the general public, as well as difficulties in diagnostics given the low sensitivity of whole-cell sonicate enzyme-linked immunosorbent assay (WCS ELISA) tests for early-stage LD [72,73,74,75]. Therefore, the date of first notification of LD may be much later than the actual first case of Lyme disease in a municipality. Another limitation of the study includes the missing dates for symptom onset (*n* = 34) and the uncertainty associated with self-reported symptom onset date, especially if several weeks or months passed since symptom onset. The location of acquisition could also be erroneous given that this is assigned by public health practitioners, whose decisions are based upon self-reported information from the Lyme disease case provided during an epidemiological investigation, and is likely subject to recall error [39]. Finally, the magnitude of speed and direction of human LD should be interpreted with caution near the edges of the study area. Estimates of speed are subject to edge effects, which indicates that estimates are less stable because they are based on fewer data (fewer neighbouring values). This edge effect can be seen in municipalities found in the sociosanitary regions of the Laurentians, Lanaudière, and Mauricie et Centre-du-Québec, where the few reported cases of human Lyme disease appear to be spreading south. As the ecology of Lyme disease is not bound by regional borders, it would be interesting to apply our methods to North America to estimate more accurate spatiotemporal patterns of Lyme disease emergence. Furthermore, a larger sample size of human LD cases would increase the stability of our estimates. Future studies could improve upon our methods by implementing more granular environmental data, including temperature, precipitation, and humidity, as well as land use changes overtime. This would increase the precision of our models not only to provide policy makers with the tools to predict when and where Lyme disease will emerge, but to also improve landscape management and reduce the risks of exposure.

## 5. Conclusions

Lyme disease continues to emerge in different regions in Québec, while increasing its presence in endemic areas. Characterizing the spatiotemporal patterns of its emergence is necessary and can be used to inform interventions, such as targeted education campaigns. Exploring the introduction of Lyme disease in Canada and globally, while identifying specific drivers of transmission, is a crucial next step. Furthermore, understanding if a surface trend approach can be used for real-time monitoring and decision aid needs further investigation.

## Figures and Tables

**Figure 1 ijerph-18-09669-f001:**
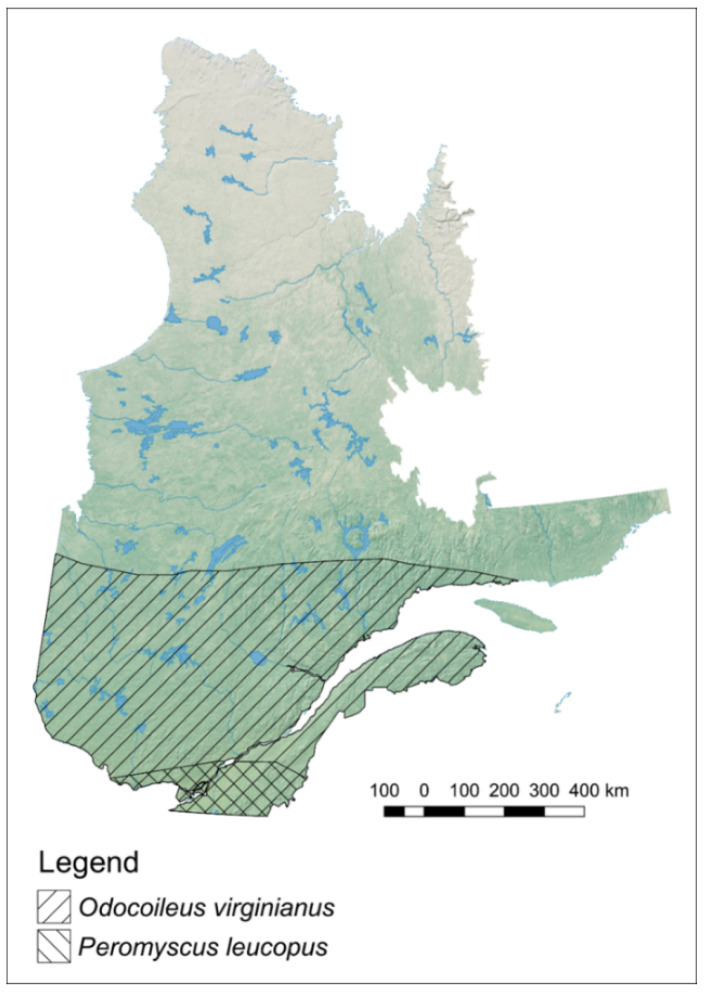
Geographical distribution of *Odocoileus virginianus* and *Peromyscus leucopus* in the province of Québec, Canada.

**Figure 2 ijerph-18-09669-f002:**
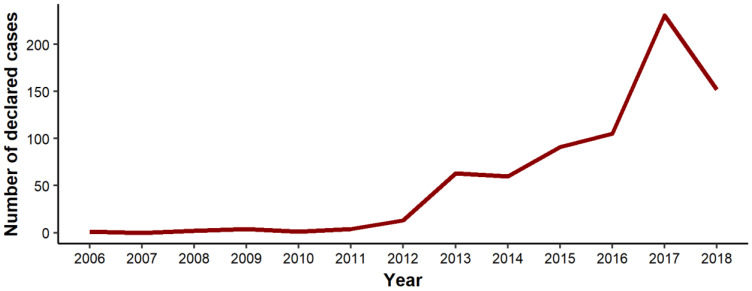
Reported Lyme disease cases in Québec, Canada from 2006 to 2018.

**Figure 3 ijerph-18-09669-f003:**
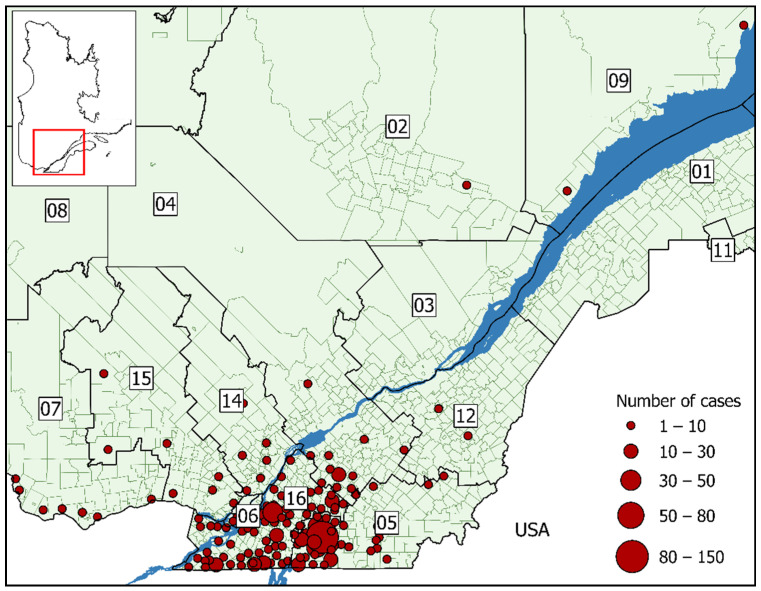
Cumulative number of Lyme disease cases in Québec acquired locally, 2006–2018 by year of notification (*n* = 727). Sociosanitary regions: (01) Bas-Saint-Laurent; (02) Saguenay-Lac-Saint-Jean; (03) Capitale-Nationale; (04) Mauricie et Centre-du-Québec; (05) Estrie; (06) Montréal; (07) Outaouais; (08) Abitibi-Témiscamingue; (09) Côte-Nord; (11) Gaspésie-Îles-de-la-Madeleine; (12) Chaudière-Appalaches; (14) Lanaudière; (15) Laurentides; (16) Montérégie.

**Figure 4 ijerph-18-09669-f004:**
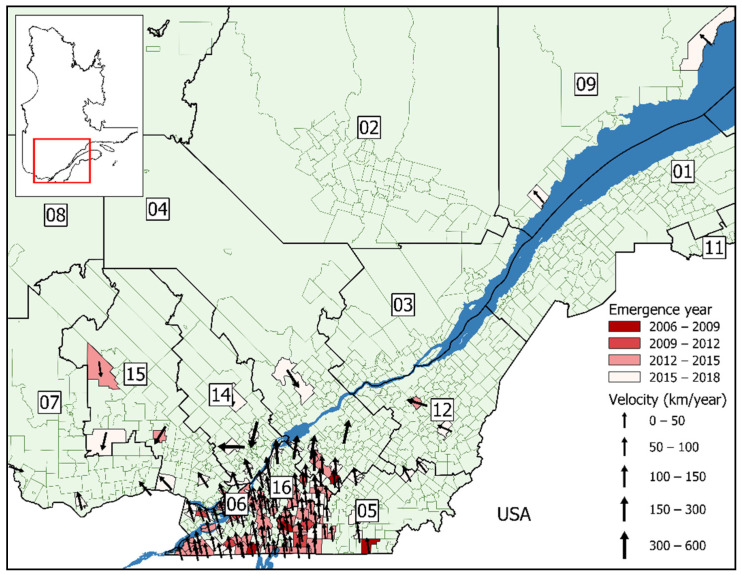
Lyme disease emergence velocity in Québec for 128 municipalities, by date of symptom onset (2006–2018). Sociosanitary regions: (01) Bas-Saint-Laurent; (02) Saguenay-Lac-Saint-Jean; (03) Capitale-Nationale; (04) Mauricie et Centre-du-Québec; (05) Estrie; (06) Montréal; (07) Outaouais; (08) Abitibi-Témiscamingue; (09) Côte-Nord; (11) Gaspésie-Îles-de-la-Madeleine; (12) Chaudière-Appalaches; (14) Lanaudière; (15) Laurentides; (16) Montérégie. Arrows indicate the direction of spread and are sized according to the speed.

**Figure 5 ijerph-18-09669-f005:**
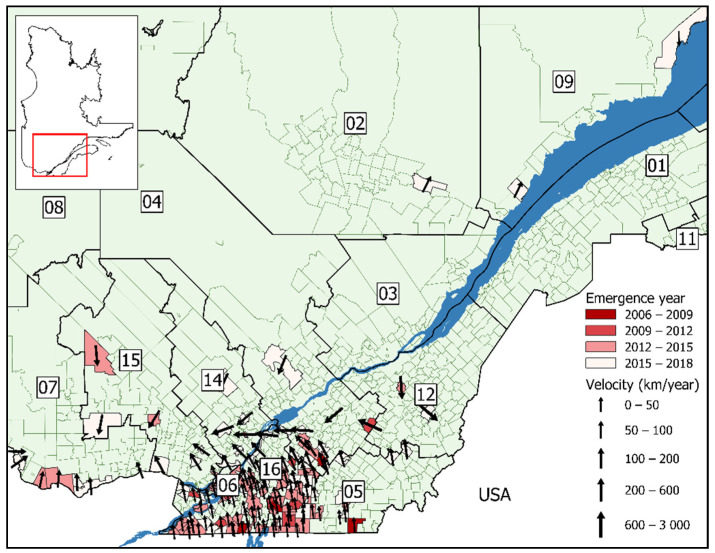
Lyme disease emergence velocity in Québec for 138 municipalities, by date of notification (2006–2018). Sociosanitary regions: (01) Bas-Saint-Laurent; (02) Saguenay-Lac-Saint-Jean; (03) Capitale-Nationale; (04) Mauricie et Centre-du-Québec; (05) Estrie; (06) Montréal; (07) Outaouais; (08) Abitibi-Témiscamingue; (09) Côte-Nord; (11) Gaspésie-Îles-de-la-Madeleine; (12) Chaudière-Appalaches; (14) Lanaudière; (15) Laurentides; (16) Montérégie. Arrows indicate the direction of spread and are sized according to the speed.

**Figure 6 ijerph-18-09669-f006:**
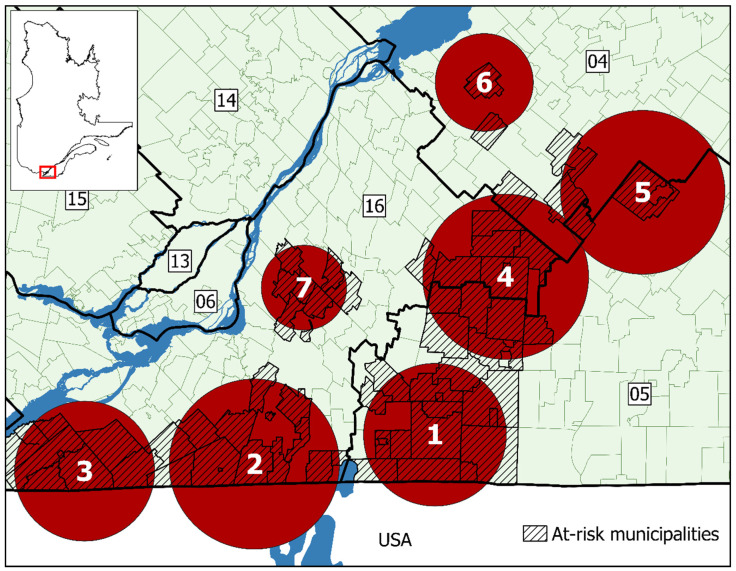
Spatiotemporal clusters of Lyme disease cases in Québec between 2006 and 2018 by date of notification (*n* = 544 cases). Clusters are shown in order of appearance. If clusters appeared in the same time period, they were numbered by decreasing count of at-risk municipalities. Each circle represents a cluster no larger than 50 km in diameter, capturing no more than 15% of the population at risk (underlying population). Sociosanitary regions: (04) Mauricie et Centre-du-Québec; (05) Estrie; (06) Montréal; (13) Laval; (14) Lanaudière; (15) Laurentides; (16) Montérégie.

**Table 1 ijerph-18-09669-t001:** Surface trend analyses based on case definition.

		Case Definition	
Scenarios		Date of Symptom Onset	Date of Notification
1	First case for each municipality between 2006–2018	A1	B1
2	Municipalities with at least two cases between 2013–2018	A2	B2
3	Municipalities with at least two cases between 2006–2018	A3	B3

**Table 2 ijerph-18-09669-t002:** Descriptive characteristics of Lyme disease cases by time period in Québec.

	All Years	2006–2012	2013–2018
Total cases	727	25	702
Number of municipalities with at least 1 case	138	20	136
Male (%)	413 (56.8%)	17 (68.0%)	396 (56.4%) *
Number of cases per age group			
1–9	77	2	75
10–19	59	2	57
20–29	49	1	48
30–39	79	3	76
40–49	81	4	77
50–59	140	5	135
60–69	171	6	165
70–79	62	2	60
80–89	9	0	9
Average time from symptom onset to medical consultation (days)	59 §	57 †	59 ‡

* Unknown sex for 2 cases; § Date of symptom onset missing for 34 cases; † Date of symptom onset missing for 1 case; ‡ Date of symptom onset missing for 33 cases.

**Table 3 ijerph-18-09669-t003:** Comparative characteristics of Lyme disease emergence in Québec between models A1 and B1, A2 and B2, and A3 and B3.

	**Model A1 ***	**Model B1 †**
Northern spread (km/year) §	32 km/year	18 km/year
Average velocity (km/year deg)	34 km/year 341° NNW	21 km/year 327° NWN
Range (km/year)	2–568 km/year	14.5–2,949 km/year
SD	67 km/year	258 km/year
	**Model A2 ****	**Model B2 ††**
Northern spread (km/year) §	40 km/year	46 km/year
Average velocity (km/year deg)	41 km/year 16° NE	47 km/year 15° NE
Range (km/year)	21–107 km/year	13–107 km/year
SD	22 km/year	19 km/year
	**Model A3 *****	**Model B3 †††**
Northern spread (km/year) §	46 km/year	44 km/year
Average velocity (km/year deg)	46 km/year 357° N	44 km/year 11° NE
Range (km/year)	13–82 km/year	12–103 km/year
SD	16 km/year	17 km/year

* Model estimating velocity of LD emergence by date of symptom onset for each municipality between 2006–2018. ** Model estimating velocity of LD emergence by date of symptom onset for municipalities with at least two cases between 2013–2018. *** Model estimating velocity of LD emergence by date of symptom onset for municipalities with at least two cases between 2006–2018. † Model estimating velocity of LD emergence by date of notification for each municipality between 2006–2018. †† Model estimating velocity of LD emergence by date of notification for municipalities with at least two cases between 2013–2018. ††† Model estimating velocity of LD emergence by date of notification for municipalities with at least two cases between 2006–2018. § Northern spread is defined as the relative velocity of emergence at 0° North.

**Table 4 ijerph-18-09669-t004:** Spatiotemporal cluster variables of Lyme disease cases in Québec between 2006 and 2018 by date of notification (*n* = 544 cases).

	Clusters						
	1	2	3	4	5	6	7
Radius (km)	20.80	24.81	20.41	23.97	23.67	14.05	12.33
Time period	2013–2018	2013–2018	2013–2018	2015–2018	2015–2018	2015–2018	2016–2018
Municipalities at-risk	15	14	8	14	4	2	7
Population at-risk	52,675	26,423	15,645	100,670	7239	2074	109,448
Declared cases	282	75	40	88	6	5	48
Expected cases	4.68	2.35	1.39	5.96	0.43	0.12	4.86
Annual cases/100,000	89.2	47.3	42.6	21.9	20.7	60.3	14.6
Relative risk	97.80	35.51	30.39	16.65	14.10	40.96	10.49
*p*-value	<0.001	<0.001	<0.01	<0.001	0.022	0.0013	<0.01

RR represents the relative risk of LD within the clusters relative to the risk outside the clusters.

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
