# Peer review of "Modelling Spatiotemporal Patterns of Lyme Disease Emergence in Québec"

_ijerph, 2021, doi:10.3390/ijerph18189669_

Round 1
Reviewer 1 Report
First of all, I would like to thank the editor for the possibility to review the manuscript entitled "Modeling spatiotemporal patterns of Lyme disease emergence in Québec." I have found it to be a good study on the epidemiology and patterns of the spread of Lyme disease in this region of Canada.
The introduction provides sufficient information and justification for conducting the study. The material and methods are correct. Despite the complexity of the analysis carried out, it is well written and understandable to the reader. The results are interesting and significant. The discussion offers an adequate vision of the studies where the current results are framed and clearly presents the limitations of the work. On the other hand, the conclusions are adequate and significant.
For all these reasons, I believe that this manuscript should be accepted for publication.
Author Response
Reviewer #1:
First of all, I would like to thank the editor for the possibility to review the manuscript entitled "Modeling spatiotemporal patterns of Lyme disease emergence in Québec." I have found it to be a good study on the epidemiology and patterns of the spread of Lyme disease in this region of Canada.
The introduction provides sufficient information and justification for conducting the study. The material and methods are correct. Despite the complexity of the analysis carried out, it is well written and understandable to the reader. The results are interesting and significant. The discussion offers an adequate vision of the studies where the current results are framed and clearly presents the limitations of the work. On the other hand, the conclusions are adequate and significant.
For all these reasons, I believe that this manuscript should be accepted for publication.
Response: Thank you very much, we appreciate the positive feedback.
Reviewer 2 Report
Dear authors,
over-all, this is a pretty good paper on an important topic. Some (mostly minor) points below.
It seems quite clear that there is hardly any relation between the reporting date and the actual course of the infection. It thus does not seem appropriate to use this data for this kind of analysis. The records of the date of symptom onset certainly have their own problems, but they appear to be the obvious choice. I'd suggest to either abandon the whole series B of models or make a much stronger point for using it in the text.
I wonder how the results would change if records from south of the borders were included, as both southern Ontario and the adjoining US states have seen quite a bit of LD as well. Neither pathogens nor ticks or birds care much about borders, so one could argue that restricting the study area in the south this way is arbitrary and will lead to inaccuracies (you already mention edge effects somewhere). Please discuss.
There are a lot of factors to consider the spread of Lyme disease. This includes occurrence and spread of not only the pathogen, but also vectors (ticks), animal hosts (mice, birds, deer, ...), and humans. Throughout the manuscript, it is often not clear to me whether you are referring to the spread of the disease as a whole (i.e. the pathogen), the occurrence of human cases, vector species, ... Please re-read carefully and adjust as necessary.
Linked with the previous point: I find it difficult to assess the status of the various links in the transmission chain throughout the study region. This is partly because of how the introduction is written: You describe the distribution of relevant species throughout Québec in broad terms. But the actual study area is only a small part of south(-eastern) Québec, and it is not clear whether the species in question do occur in the study region (see also comment on ll 208-209 below), and if yes, where and how commonly. An additional map (even a rough, hand-drawn sketch) of their approximate ranges would be very helpful here. Or at least some additional sentences in the introduction that explain the occurrence of the relevant species throughout the area shown in the figures.
l 25: I do not understand the term "rate of risk". "rate" suggests change, so maybe something like "high rate of increase of disease risk" is meant?
l 93: I assume "date of reporting" here means the same as "date of notification" elsewhere in the text? If yes: stick to one term, otherwise explain the difference.
l 123: what does the acronym INSPQ stand for?
l 149: please elaborate: how was the relative risk calculated?
ll 223-224: What is "with the purely spatial clusters" supposed to mean?
ll 247-265: This whole first paragraph of the discussion feels out of place, as it is about results that are not relevant to the main objective of the paper (spatio-temporal patterns). Suggest to move it further downwards and start the discussion with the following paragraph(s).
ll 308-309: I fear you'll have to elaborate on that a bit: human infections spreading faster than the vectors does not seem plausible at first sight - unless it is spreading into areas where the vectors are already present.
Figures in general: I like them! Almost nothing for me to complain about, and that's a rare thing – well done.
Figures 3 & 4 plus similar ones in the appendix: the different arrow sizes are difficult to distinguish in the maps. Maybe 3 or four different sizes would be enough?
All supplementary figures: what do the numbers in squares mean? -> add sociosanitary regions to descriptions.
Figure S7 and following: What do the numbers in the circles mean? What does the size of the circles mean? -> repeat explanation from fig 5.
Author Response
Reviewer #2:
Dear authors,
Over-all, this is a pretty good paper on an important topic. Some (mostly minor) points below.
Comment #1
It seems quite clear that there is hardly any relation between the reporting date and the actual course of the infection. It thus does not seem appropriate to use this data for this kind of analysis. The records of the date of symptom onset certainly have their own problems, but they appear to be the obvious choice. I'd suggest to either abandon the whole series B of models or make a much stronger point for using it in the text.
Response #1
We agree in theory with the reviewer about this point on date of symptom onset although it is very problematic for Lyme disease, given the ambiguity of the symptoms particularly for those with early disseminated or late disseminated LD. Furthermore, the time lapse from when they were exposed to when they report their symptom onset date (during the case investigation with public health), could be several weeks to several months. We have included a sentence in the manuscript as to why we have chosen to include both dates (line 127-128).
Comment #2
I wonder how the results would change if records from south of the borders were included, as both southern Ontario and the adjoining US states have seen quite a bit of LD as well. Neither pathogens nor ticks or birds care much about borders, so one could argue that restricting the study area in the south this way is arbitrary and will lead to inaccuracies (you already mention edge effects somewhere). Please discuss.
Response #2
Line 382-384: We have added a sentence about this limitation in the discussion.
Comment #3
There are a lot of factors to consider the spread of Lyme disease. This includes occurrence and spread of not only the pathogen, but also vectors (ticks), animal hosts (mice, birds, deer, ...), and humans. Throughout the manuscript, it is often not clear to me whether you are referring to the spread of the disease as a whole (i.e. the pathogen), the occurrence of human cases, vector species, ... Please re-read carefully and adjust as necessary.
Response #3
We have adjusted the text throughout the manuscript to better differentiate spread between human, the pathogen, ticks, and reservoirs.
Comment #4
Linked with the previous point: I find it difficult to assess the status of the various links in the transmission chain throughout the study region. This is partly because of how the introduction is written: You describe the distribution of relevant species throughout Québec in broad terms. But the actual study area is only a small part of south(-eastern) Québec, and it is not clear whether the species in question do occur in the study region (see also comment on ll 208-209 below), and if yes, where and how commonly. An additional map (even a rough, hand-drawn sketch) of their approximate ranges would be very helpful here. Or at least some additional sentences in the introduction that explain the occurrence of the relevant species throughout the area shown in the figures.
Response #4
We have added Figure 1 on page 2 to depict the geographical distribution of the white-footed mouse and the white-tailed deer.
Comment #5
l 25: I do not understand the term "rate of risk". "rate" suggests change, so maybe something like "high rate of increase of disease risk" is meant?
Response #5
Line 28: This was changed to “A significantly high risk of disease”.
Comment #6
l 93: I assume "date of reporting" here means the same as "date of notification" elsewhere in the text? If yes: stick to one term, otherwise explain the difference.
Response #6
Line 99: We have changed this to “date of notification” throughout the manuscript.
Comment #7
l 123: what does the acronym INSPQ stand for?
Response #7
The acronym INSPQ is first described on Line 92. It is French for Institut National de Santé Publique du Québec. It has been translated in English as Québec National Institute of Public Health.
Comment #8
l 149: please elaborate: how was the relative risk calculated?
Response #8
We have added this to the manuscript:
Line 157-163: The RR is calculated by dividing the observed cases with the expected cases within the cluster, the result of which is then divided with the observed cases divided by the expected cases outside the cluster, as represented by the following formula:
where c=observed cases within the cluster, E[c]=expected number of cases, and C = total number of cases in the dataset.
Comment #9
ll 223-224: What is "with the purely spatial clusters" supposed to mean?
Response #9
Line 239: It was an incomplete sentence, which now reads: “….with the purely spatial clusters presented in the supplementary file.”
Comment #10
ll 247-265: This whole first paragraph of the discussion feels out of place, as it is about results that are not relevant to the main objective of the paper (spatio-temporal patterns). Suggest to move it further downwards and start the discussion with the following paragraph(s).
Response #10
Line 343-362: This paragraph was moved after discussion on LD emergence and clusters.
Comment #11
ll 308-309: I fear you'll have to elaborate on that a bit: human infections spreading faster than the vectors does not seem plausible at first sight - unless it is spreading into areas where the vectors are already present.
Response #11
Line 333-339: We have added a section to further discuss how this is may be plausible. Human cases may appear in isolated municipalities due to the role of migratory birds, before tick-carrying mice or deer are seen in these regions. Isolated municipalities far from one another also skew the speed of emergence towards faster spread.
Comment #12
Figures in general: I like them! Almost nothing for me to complain about, and that's a rare thing – well done.
Response #12
Thank you very much!
Comment #13
Figures 3 & 4 plus similar ones in the appendix: the different arrow sizes are difficult to distinguish in the maps. Maybe 3 or four different sizes would be enough?
Response #13
The number of arrow sizes in Figures S4-S8 were reduced to three. Arrow sizes for all maps were adjusted to better distinguish them.
Comment #14
All supplementary figures: what do the numbers in squares mean? -> add sociosanitary regions to descriptions.
Response #14
Sociosanitary regions were added to supplementary figures
Comment #15
Figure S7 and following: What do the numbers in the circles mean? What does the size of the circles mean? -> repeat explanation from fig 5.
Response #15
The description of supplemental Figures S7 and S8 were adjusted to explain cluster numbering and cluster size. The explanation from Figure 5 was copied.
Reviewer 3 Report
Dear Authors,
Thank you very much for allowing me to review your manuscript. I found it very interesting, and I only have a few minor comments and suggestions:
- On page 1, line 39, remove "is" from "making LS is a significant..."
- On page 2, line 93, replace "case" with ",such as ".
- On page 3, line 123, you mention the INSPQ definition of Lyme disease risk. What does INSPQ mean? Can you please specify?
- On page 10, lines 247-249, there's a discrepancy between what is reported in the Results and Table 1, and what is reported here. Here, you talk about a bimodal distribution with 5-14 and 50-69 year old age groups at more risk; however, this is not mentioned in the Results, and in Table 1, you show 1-9,10-19, 50-59 and 60-69 year olds. Please make these match up!
- On page 10, line 248, you say that "men were also statistically more at-risk than women, particularly those aged 15 to 34 years". However, I have not seen any statistical test results in the Results section, with p-values. Please include those in the results, or remove the word "statistically" from this sentence. Also, the age range here does not match the age range in Table 1.
- On page 10, line 250, you state that "35 and 60 year olds are at high risk". Why those exact ages? How about 36 and 61 year olds? I'm sure there's a miscommunication here, please correct.
- On page 10, line 256, please insert a comma after "However".
- On page 11, line 332, you state that from some regions "LD seems to be spreading south". Why would that be, what would be the mechanism for that? Is it possible that those results are statistical anomalies, just basically erroneous estimates, due to cases reported in isolated locations, or perhaps not where the cases were acquired.
- Finally, I have a suggestion. You describe in the discussion a number of mechanisms that could explain the northward spread of Lyme disease, including both climate change and land-use land-cover change. Now that you have estimates for the speed and direction of spread, it would be interesting to try to correlate the changes in temperature and climate, as well as the amount of land-use-land-cover change in each of the counties, so that you could see how much temperature or LULC change is driving this spread.
Thank you very much again for the opportunity to read your work, I will certainly mention it in my classes on climate change and disease ecology!
Author Response
Reviewer #3:
Dear Authors,
Thank you very much for allowing me to review your manuscript. I found it very interesting, and I only have a few minor comments and suggestions:
Comment #1
On page 1, line 39, remove "is" from "making LS is a significant..."
Response #1
Line 43: The word “is” was removed from “making LS is a significant…”
Comment #2
On page 2, line 93, replace "case" with ",such as ".
Response #2
Line 99: “case” was replaced with “, such as”
Comment #3
On page 3, line 123, you mention the INSPQ definition of Lyme disease risk. What does INSPQ mean? Can you please specify?
Response #3
The acronym INSPQ is first described on Line 92. It is French for Institut National de Santé Publique du Québec. It has been translated in English as Québec National Institute of Public Health.
Comment #4
On page 10, lines 247-249, there's a discrepancy between what is reported in the Results and Table 1, and what is reported here. Here, you talk about a bimodal distribution with 5-14 and 50-69 year old age groups at more risk; however, this is not mentioned in the Results, and in Table 1, you show 1-9,10-19, 50-59 and 60-69 year olds. Please make these match up!
Response #4
Line 179-183: Adjusted wording to mention bimodal distribution of cases found in age groups 1-19 and 50-69.
Line 343: Adjusted age group from 5-14 to 1-19.
Comment #5
On page 10, line 248, you say that "men were also statistically more at-risk than women, particularly those aged 15 to 34 years". However, I have not seen any statistical test results in the Results section, with p-values. Please include those in the results, or remove the word "statistically" from this sentence. Also, the age range here does not match the age range in Table 1.
Response #5
Line 179-183: We have added a statistically significant result for men being at higher risk than women. RR = 1.34 [CI95: 1.16 – 1.55]).
Comment #6
On page 10, line 250, you state that "35 and 60 year olds are at high risk". Why those exact ages? How about 36 and 61 year olds? I'm sure there's a miscommunication here, please correct.
Response #6
Line 346-347: Adjusted to “A similar demographic trend has been documented in the United States, where cases of Lyme disease are more prevalent among men aged 5-15 years and 45-55 years”.
Comment #7
On page 10, line 256, please insert a comma after "However".
Response #7
Line 353: Comma inserted after “However”.
Comment #8
On page 11, line 332, you state that from some regions "LD seems to be spreading south". Why would that be, what would be the mechanism for that? Is it possible that those results are statistical anomalies, just basically erroneous estimates, due to cases reported in isolated locations, or perhaps not where the cases were acquired.
Response #8
Lines 379-382: Clarified that this may be due to edge effect, where estimates are less accurate as they are based on fewer data.
Comment #9
Finally, I have a suggestion. You describe in the discussion a number of mechanisms that could explain the northward spread of Lyme disease, including both climate change and land-use land-cover change. Now that you have estimates for the speed and direction of spread, it would be interesting to try to correlate the changes in temperature and climate, as well as the amount of land-use-land-cover change in each of the counties, so that you could see how much temperature or LULC change is driving this spread.
Response #9
Thank you for your suggestion. We have added a few sentences on this topic in the discussion in Lines 385-390.
Comment #10
Thank you very much again for the opportunity to read your work, I will certainly mention it in my classes on climate change and disease ecology!
Response #10
Thank you very much!